# Use of a paediatric advice line for parents of infants recruited to a randomised controlled trial

Joyce Chan [1,2] Jordan Kirby [3] Kaya Kareela Gardiner,[1] Veronica Abruzzo,[1] Samantha Bannister,[1,4] Karen Bellamy,[5,6] Clare Brophy,[7,8] Vanessa Clifford,[4,9] Lianne Cox,[7,10] Hannah Elborough,[7] Bridget Freyne Freyne,[11] Freya Summons,[1,12] Clare Morrison,[7] Kate Wall,[7] Daniel M Casalaz,[13] Nigel Curtis [1,12] Laure F Pittet,[1,12] on behalf of the MIS BAIR trial group

JC and JK are joint first authors. NC and LFP are joint senior authors.

For numbered affiliations see end of article.

**Correspondence to**
Dr Joyce Chan; joyce.chan@mcri.edu.au

## ABSTRACT

**Background** This study aims to describe the use of a paediatric advice line (PAL) provided to parents whose infants were recruited to a large randomised controlled trial (RCT), including the number and types of medical concerns addressed, seasonal variability and call outcomes. Additionally, sociodemographic characteristics of the parents and children of those parents who used the PAL are compared with those who did not.

**Methods** Prospective cohort of 1246 children nested in the Melbourne Infant Study: BCG for Allergy and Infection Reduction (MIS BAIR) RCT. All MIS BAIR participants were offered access to the PAL. Data were collected over the initial 2 years of a 5-year follow-up. Data were analysed using $\chi^2$ tests, and ORs were calculated using multiple logistic regression.

**Results** The PAL was used by 230 (18.5%) participants, who made a total of 586 calls during the 2-year study period. The reasons for calling the PAL were dermatological (24%); gastrointestinal (18%); disturbances in feeding, sleeping and crying (14%); respiratory (7%); and developmental/neurological (6%). Analysis revealed that those who used the PAL were more likely to be first-time parents (OR 1.4, 95% CI 1.1 to 1.9) and mothers who hold a university degree (OR 3.3, 95% CI 1.3 to 8.4). PAL costs were minimal and comprised 15 clinicians with paediatric experience.

**Conclusions** A cost-effective PAL service for clinical trial participants was used appropriately by parents for relatively minor concerns and may have a role in trials to promote participant engagement and reduce demand for other health services.

### WHAT IS ALREADY KNOWN ON THIS TOPIC

⇒ Paediatric advice services improve patient satisfaction, help-seeking and informed decision making.
⇒ Medical advice hotlines have yet to be studied in paediatric clinical trials.

### WHAT THIS STUDY ADDS

⇒ Parental demand for non-urgent advice within a clinical trial exists for a variety of medical concerns.
⇒ First-time parents and mothers with a university degree were more likely to use the paediatric advice line.

### HOW THIS STUDY MIGHT AFFECT RESEARCH, PRACTICE OR POLICY

⇒ Advice lines for non-urgent medical concerns offer an economical and valuable means of parental engagement in paediatric clinical trials.

PAL study, a prospective cohort study, was embedded within the MIS BAIR RCT. The intention was to provide access to non-urgent, telephone-based, paediatric medical advice. Here we report the use of the MIS BAIR PAL, including the number and types of medical concerns addressed, seasonal variability, call outcomes and the set-up of the PAL. Additionally, we describe the sociodemographic characteristics of those parents who used the PAL compared with those who did not, as well as antenatal, intrapartum and early parenthood factors.

## METHODS

### Study setting and PAL development

In the MIS BAIR trial, 8552 participants were assessed for eligibility with 7280 excluded and 1272 healthy neonates randomised to receive BCG-Denmark vaccine or no vaccine within 10 days of birth (figure 1). The MIS BAIR eligibility criteria are published

## INTRODUCTION

The Melbourne Infant Study: BCG for Allergy and Infection Reduction (MIS BAIR) is an ongoing, multicentre randomised controlled trial (RCT) to determine the off-target effects of neonatal BCG vaccination on allergies, eczema, infections and asthma in Australian infants.[1] To provide some benefit to parents of participants of the trial, we established a paediatric advice line (PAL). This MIS BAIR

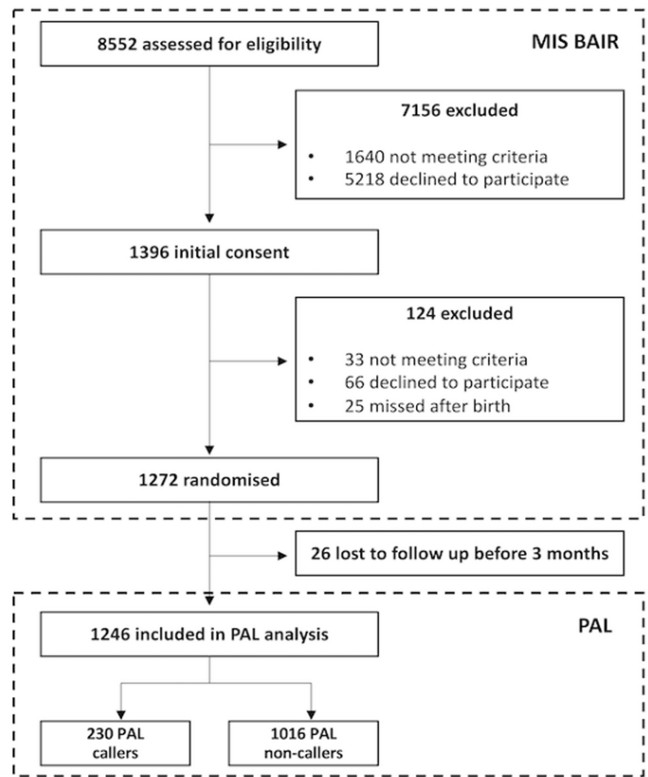

**Figure 1** CONSORT diagram for MIS BAIR PAL study. MIS BAIR, Melbourne Infant Study: BCG for Allergy and Infection Reduction; PAL, paediatric advice line.

open-access in the MIS BAIR study protocol.[1] Infants were followed up regularly for up to 5 years with parent-completed questionnaires and blood sample collection.[1] A PAL was made available to the parents of all the infants recruited to MIS BAIR for the first 2 years of their participation as recognition for their contribution to the trial.

The PAL involved a single contact number linked to a mobile phone carried by a clinical staff member during standard business hours (0900–1700 on weekdays). Parents were told that it was intended only for non-urgent medical questions and a voicemail could be left by callers outside of standard business hours with a return call within one business day. A clinical member of the study team (doctor, nurse or midwife with paediatric experience) responded to all calls and provided advice (sometimes with emailed resources). The PAL comprised 15 clinicians rostered to hold the PAL mobile device on a voluntary rotating roster for the duration of one day. If a nurse or midwife receiving PAL calls on a particular day required medical input, a referral was completed to a medical MIS BAIR staff member with a return call within one business day. Parents calling for a rash or skin condition were asked to send photographs by email or smartphone. If appropriate, follow-up calls were arranged with a nurse, a paediatrician, the participant's general practitioner, a specialist clinic or at the emergency department.

### Data collection

Details of all calls were recorded by the clinical member answering the call using a Research Electronic Data Capture (REDCap) database, including principal concerns and outcome of the call. Follow-up calls were recorded in chronological order and linked to the same participant number. The data were subsequently cleaned, categorised and coded in a standardised manner by JC and JK.

### Definitions

A PAL caller was defined as a parent who remained in the trial for at least 3 months (completed at least one of the 3-monthly questionnaires) and who called the PAL at least once during the first 2 years of their participation in the trial. A PAL non-caller was defined as a parent who remained in the trial for at least 3 months (completed at least one of the 3-monthly questionnaires) but who did not call the PAL. Parents who called exclusively with questions about the trial (including BCG vaccine injection site concerns) were defined as non-callers, and these calls were not included in the analysis. The principal concern was determined as the initial symptom discussed during the call, with other symptoms listed as additional concerns.

### Statistical analysis

The sociodemographic characteristics of PAL callers were compared with non-callers using $\chi^2$ tests, Fisher's exact tests, independent samples t-tests and Wilcoxon-Mann-Whitney tests. Univariate logistic regression was used to identify independent factors associated with the use of the PAL. All significant predictors (p<0.20) were then included in the final multiple logistic regression, with a stepwise backward exclusion of factors with p values of >0.05. Statistical analysis was performed using Stata V.16.

### Patient and public involvement

The MIS BAIR PAL was developed with the interests of the public and the MIS BAIR participants at the forefront of the study design. The primary outcome was to develop a PAL and observe parental use with the aim of improving patient and parental engagement. Patients were not directly involved in the development of the study design, and future analyses aim to further investigate parental experience and satisfaction with the PAL.

### RESULTS

A total of 1246 participants were included in the final analysis and 26 were lost to follow-up after randomisation (2.0%) (figure 1). The demographics of the study population are detailed in table 1. There was an approximately even distribution of sex of the infant and their BCG vaccination status. Most participants were born via vaginal delivery, and most were breast fed.

A total of 586 calls were recorded during the study. A total of 230 parents (18.5%) called the PAL at least once

**Table 1** PAL study parent and infant demographics

| | All participants (N=1246) | PAL callers (n=230) | PAL non-callers (n=1016) | P value |
|---|---|---|---|---|
| **Sex of child** | | | | |
| Male | 628 (50.4) | 117 (50.9) | 511 (50.3) | 0.9 |
| Female | 618 (49.6) | 113 (49.1) | 505 (49.7) | |
| **First-time parent** | | | | |
| Yes | 649 (52.1) | 140 (60.9) | 509 (50.1) | 0.003 |
| No | 597 (47.9) | 90 (39.1) | 507 (49.9) | |
| **Number of siblings** | | | | |
| 0 | 649 (52.1) | 140 (60.9) | 509 (50.1) | 0.02 |
| 1 | 388 (31.1) | 69 (30.0) | 319 (31.4) | |
| 2 | 150 (12.0) | 15 (6.5) | 135 (13.3) | |
| 3 | 34 (2.7) | 4 (1.7) | 30 (3.0) | |
| 4 | 13 (1.0) | 1 (0.4) | 12 (1.2) | |
| 5 | 10 (0.8) | 1 (0.4) | 9 (0.9) | |
| 6 | 0 (0.0) | 0 (0.0) | 0 (0.0) | |
| 7 | 2 (0.2) | 0 (0.0) | 2 (0.2) | |
| **Birth method** | | | | |
| Vaginal | 800 (64.2) | 150 (65.2) | 650 (64.0) | 0.7 |
| Caesarean | 446 (35.8) | 80 (34.8) | 366 (36.0) | |
| **Gestational age** | | | | |
| Weeks* | 39.3 (38.4, 40.3) | 39.4 (38.5, 40.5) | 39.3 (38.4, 40.3) | 0.1 |
| **In-vitro fertilisation (IVF) or sperm/egg donor** | | | | |
| IVF/donor used | 20 (1.6) | 8 (3.5) | 12 (1.2) | 0.003 |
| No IVF/donor used | 333 (26.7) | 49 (21.3) | 284 (28.0) | |
| Unknown | 893 (71.7) | 173 (75.2) | 720 (70.9) | |
| **Birth weight** | | | | |
| Birth weight (kg)* | 3.4 (3.1, 3.7) | 3.4 (3.1, 3.8) | 3.4 (3.1, 3.7) | 0.3 |
| **Maternal age at delivery** | | | | |
| Years† | 32.7 (4.7) | 33.3 (4.1) | 32.6 (4.9) | 0.06 |
| **Mother's birthplace** | | | | |
| Australia | 902 (72.4) | 169 (73.5) | 733 (72.2) | 0.2 |
| Outside Australia | 344 (27.6) | 61 (26.5) | 283 (27.9) | |
| **Maternal education** | | | | |
| No education/up to year 10 | 69 (5.5) | 5 (2.2) | 64 (6.3) | <0.001 |
| Year 12/trade | 326 (26.2) | 39 (17.0) | 287 (28.1) | |
| University | 848 (68.1) | 185 (80.4) | 663 (65.3) | |
| Unknown | 3 (0.2) | 1 (0.4) | 2 (0.2) | |
| **Breast fed** | | | | |
| Breast fed | 927 (74.4) | 187 (81.3) | 740 (72.8) | 0.03 |
| Not breast fed | 37 (3.0) | 2 (0.9) | 35 (3.4) | |
| Unknown | 282 (22.6) | 41 (17.8) | 241 (23.7) | |
| **Received BCG** | | | | |
| Yes | 622 (49.9) | 119 (51.7) | 503 (49.5) | 0.4 |
| No | 624 (50.1) | 111 (48.3) | 513 (50.5) | |

Continued

**Table 1** Continued

| | All participants (N=1246) | PAL callers (n=230) | PAL non-callers (n=1016) | P value |
|---|---|---|---|---|
| Reported as number (%) and p value calculated using $\chi^2$ or Fisher's exact test, unless otherwise noted. *Reported as median (IQR), p value calculated using Wilcoxon-Mann-Whitney test. †Reported as mean (SD), p value calculated using independent samples t-test. PAL, paediatric advice line. | | | | |

during the first 2 years of their participation in the trial. Of these, 69 (30%) contacted the PAL on only one occasion (inclusive of subsequent follow-up calls) and 161 (70%) on multiple occasions.

## Reasons for call

Figures 2 and 3 illustrate the age distribution and the principal concerns for the PAL calls. The most common reason for a call was a rash or skin condition (87 calls, 24.5%), with calls distributed equally across infancy. The second most common reason was for a concern relating to feeding, sleeping or crying (49 calls, 14.6%), with most occurring before 20 weeks of age. A total of 34 (10.1%) and 30 calls (9.0%) were related to changes in bowel habit or vomiting, respectively, with most calls received during the neonatal period. Calls related to respiratory symptoms (26 calls, 7.8%) had a bimodal presentation, clustered around 10 and 60 weeks of age. There were 8 calls (2.4%) relating to concerns about lumps/bumps/masses, all received in the first year of life. A total of 12 calls (3.6%) were for fever. Of the 87 dermatologically related calls, 3 (0.9%) were associated with a fever. Of the 26 respiratory symptom-related calls, 4 (1.2%) were associated with a fever. Developmental or neurological concerns (21 calls, 6.3%) occurred in early infancy, as did growth concerns (4 calls, 1.2%). Calls listed as 'other medical concern—baby unwell' included 32 calls (9.6%) for various concerns including ear/nose/throat issues (12 calls, 3.6%), abnormal pathology results or medication queries. A total of 37 calls (11.0%) were for general advice, mainly related to travel (9 calls, 2.7%), obtaining vaccines (12 calls, 3.6%) and introduction of solids to the infant's diet (12 calls, 3.6%).

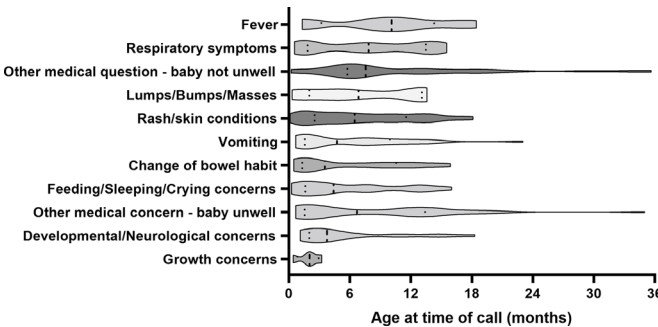

**Figure 2** Age distribution of infants, in months, at the time of the parents' call to the paediatric advice line categorised by reasons.

## Seasonal variability

Reasons for PAL calls showed seasonal variability (figure 4). Calls occurred most frequently during winter (31.5% of all calls), followed by autumn (24.5%), spring (22.3%) and least frequently during summer (21.7%). There was a larger number of calls for rashes and other skin conditions during winter (31.7% of all calls regarding a rash) and summer (29.3%), compared with during autumn (20.7%) and spring (18.3%). Calls related to feeding, sleeping, crying, changes of bowel habit, vomiting or respiratory symptoms were more numerous during autumn and winter. Travel advice was sought most commonly during winter. However, variability between seasons for all reasons for PAL calls were statistically

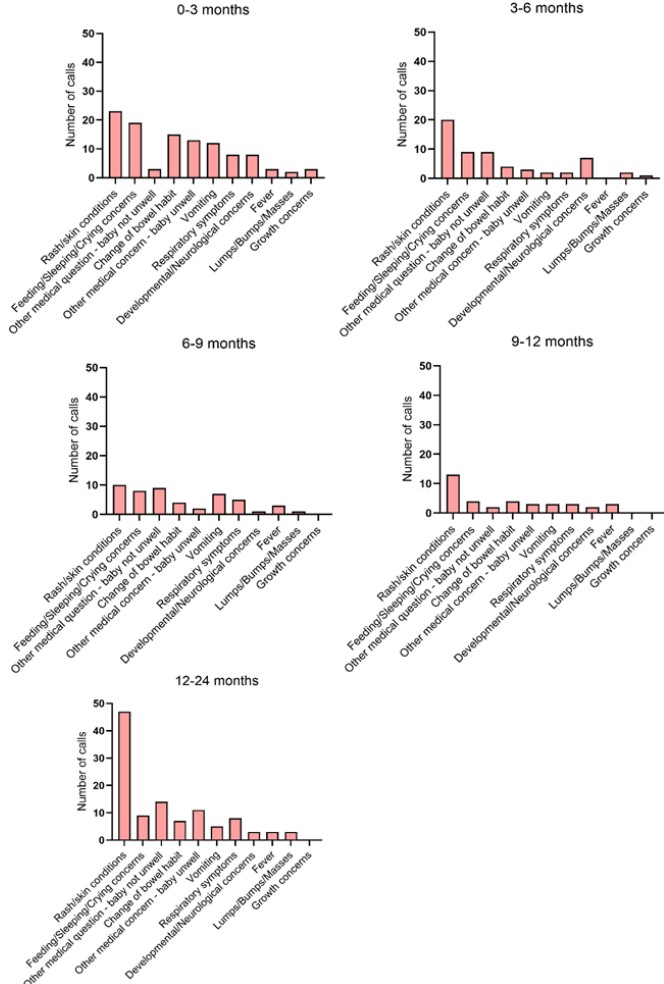

**Figure 3** Frequency of paediatric advice line calls for each reason at different infant ages.

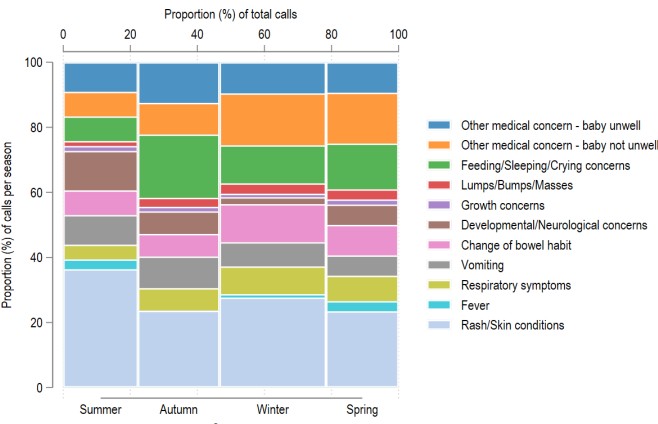

**Figure 4** Seasonal distribution of calls to the paediatric advice line for each reason during the first 2 years of life. Season and the proportion of calls for each season are displayed on the x-axis. The proportion of calls for each reason within a specific season is displayed along the y-axis.

insignificant on $\chi^2$ testing of proportions. Participant recruitment was approximately evenly distributed across all seasons, accounted for as a potential confounding factor, as shown in online supplemental appendix A.

### Call outcomes

Of the 586 calls to the PAL, 314 (53.6%) of the parents' concerns were resolved in a single call, 185 (31.7%) in two calls and 87 (14.8%) in three or more calls. For the 268 calls for which this information was available, the outcomes were verbal medical advice given by a paediatrician (141 calls, 52.6%), verbal advice from a paediatric nurse or midwife (60 calls, 22.4%), written advice from a PAL clinician (email or text message) (18 calls, 6.7%), referral to be seen in-person by a paediatrician (11 calls, 4.1%), and referral to a general practitioner (31 calls, 11.6%) or the emergency department (seven calls, 2.6%). Advice was either given during the initial call or shortly after if advice from a more senior clinician was required.

### Sociodemographics of PAL callers

The sociodemographic characteristics of PAL callers and non-callers were compared (table 1). PAL callers had a slightly higher maternal age (mean 33.3 years, SD 4.1) compared with non-callers (mean 32.6 years, SD 4.9, p=0.06, t-test). First-time parents were more likely to call the PAL than non-first-time parents (140/649 (21.6%) vs 90/597 (15.1%), p=0.003, $\chi^2$ test). Parents of infants conceived though IVF or donor gametes were also more likely to call (8/20 (40%) vs 49/333 (14.7%), p=0.003, $\chi^2$ test), as were parents of infants who were breast fed (187/927 (20.2%) vs 2/37 (5.4%), p=0.03, Fisher's exact test) and whose mothers were university-educated (185/848 (22%) vs 44/395 (11%), p<0.001, $\chi^2$ test). There were no statistically significant differences between the PAL caller and non-caller groups in relation to sex of the infant, mode of delivery, birth weight, maternal birthplace or BCG vaccination status.

### Multivariate analysis

Univariate and multiple logistic regression analyses are outlined in table 2. In the multiple logistic regression analysis, being a first-time parent remained a statistically significant predictor for calling the PAL (OR 1.4, 95% CI 1.1 to 1.9), as did maternal attainment of a university degree (OR 3.3, 95% CI 1.3 to 8.4). Breast feeding and higher maternal age were not retained in the final multivariate analysis as they were no longer predicting the outcome, likely due to them being confounded by higher maternal education status in the model.

### DISCUSSION

Paediatric advice services provide guidance for parents, enhance clinician–patient communication, and improve parental satisfaction and accessibility to non-urgent health advice.[2–4] Given approximately one in five parents used the PAL, with a high proportion of parents calling multiple times, this highlights a demand for non-urgent medical advice during early childhood. While studies have shown a growing demand for medical triage and assessment hotlines, this has not been studied in a clinical trial setting.[5]

The PAL operated via a single mobile phone attended by a clinical staff member during standard business hours on a voluntary rotating roster basis. The overall cost was minimal, with one-off costs for a mobile device and telephone number, given the voluntary basis of the team and the operation during business hours. Given 46.4% of PAL calls required a follow-up consultation, continuity of care was one challenge faced by PAL clinicians. However, detailed clinical documentation with all follow-up consultations occurring within one business day allowed for a streamlined process. Telephone consultation without real-time visualisation of a dermatological complaint provided some difficulty in clinical assessment. Photographs provided by parents assisted in clinical decision making. However, using a live audiovisual telehealth service in future studies may ameliorate this challenge.

In our study, the majority of concerns could be addressed by PAL staff alone, and the large majority of concerns were resolved within two telephone calls. This is likely to have reduced the demand on emergency services for non-urgent concerns. Factors including the parent's perception of urgency, uncertainty in their own assessment of the child's symptoms and unawareness of service availability all influence a parent's decision to seek help.[6] Providing a PAL for non-urgent concerns provides an accessible means for parents to understand the importance of their child's symptoms and better inform parental decision making in relation to whether they need to seek further help.

Dermatological concerns were the most common reason to contact the PAL across all ages, with a proclivity for summer and winter months. Many dermatological conditions, such as miliaria, atopic dermatitis and viral exanthems, present frequently during early childhood,

**Table 2** Univariate and multiple logistic regression analyses predicting PAL callers

| | Univariate model | | Multivariate model | |
| --- | --- | --- | --- | --- |
| | OR* (95% CI) | P-value | OR* (95% CI) | P value |
| **Sex** | | | | |
| Male | 1.0 (0.8 to 1.4) | 0.9 | | |
| **First-time parent** | | | | |
| Yes | 1.5 (1.2 to 2.1) | 0.003 | 1.4 (1.1 to 1.9) | 0.02 |
| **Number of siblings** | | | | |
| 0 | Reference | Reference | | |
| 1 | 0.8 (0.6 to 1.1) | 0.1 | | |
| 2 | 0.4 (0.2 to 0.7) | 0.002 | | |
| 3 | 0.5 (0.2 to 1.4) | 0.2 | | |
| 4 | 0.3 (0.04 to 2.4) | 0.3 | | |
| 5 | 0.4 (0.05 to 3.2) | 0.4 | | |
| **Birth method** | | | | |
| Vaginal | 1.1 (0.8 to 1.4) | 0.7 | | |
| **Gestational age** | | | | |
| Weeks | 1.1 (1.0 to 1.2) | 0.08 | | |
| **IVF or sperm/egg donor** | | | | |
| Yes | 3.9 (1.5 to 9.9) | 0.005† | | |
| **Birth weight** | | | | |
| <2500 g | 0.8 (0.3 to 1.8) | 0.5 | | |
| Birth weight (kg) | 1.2 (0.9 to 1.6) | 0.2 | | |
| **Maternal age at delivery** | | | | |
| Years | 1.0 (1.0 to 1.1) | 0.06 | | |
| **Mother's birthplace** | | | | |
| Australia | 1.1 (0.8 to 1.5) | 0.7 | | |
| **Maternal education** | | | | |
| No education/up to year 10 | Reference | Reference | Reference | |
| Year 12/trade | 1.7 (0.7 to 4.6) | 0.3 | 1.7 (0.6 to 4.5) | 0.3 |
| University | 3.6 (1.4 to 9.0) | 0.007 | 3.3 (1.3 to 8.4) | 0.01 |
| **Breast fed** | | | | |
| Yes | 4.4 (1.1 to 18.6) | 0.04 | | |
| **Received BCG** | | | | |
| Yes | 1.1 (0.8 to 1.5) | 0.5 | | |

*OR calculated using logistic regression method.
†IVF was not included in the multivariate analysis due to missing data resulting in loss of statistical power.
PAL, paediatric advice line.

often with seasonal variability.[7–9] Calls for respiratory concerns, gastrointestinal symptoms and disturbances with sleeping, feeding and crying were all clustered within the first 20 weeks of life; this is most likely due to early parenthood anxiety, given the high prevalence of first-time and breastfeeding parents among the PAL callers. The increased frequency of these calls during the winter months most likely correlates with higher incidence rates of bronchiolitis, croup and viral infections, with associated gastrointestinal upset and irritability.[10 11] The clustering of 'other' medical concerns in well infants around

30 weeks of age may correspond with the introduction of solids into the diet, often a time of apprehension.[12] Only a small proportion of calls were related to febrile episodes, even accounting for fever listed as an associated symptom rather than the main presenting concern. This may be due to parents presenting to a medical setting directly, recognising it as a potential emergency.[13]

PAL callers were more likely to be older mothers, first-time parents, university-educated and/or have used IVF or donor gametes compared with non-callers. This is consistent with literature indicating that first-time

mothers of a higher maternal age have increased levels of psychological stress and anxiety during both pregnancy and the postpartum period.[14] Pregnancies acquired with the use of IVF and oocyte/sperm donation are more likely to develop stress and anxiety,[15] partly as a result of the greater risk of antenatal and intrapartum complications.[16 17] Mothers with a university education might also feel more empowered to ask for advice than those without or those with a language barrier. Further investigation of the influence of other socioeconomic factors, such as health literacy, income and employment status, on the willingness to access and use the advice line is required as this study suggests lower educational attainment may be a barrier to PAL use.

While patient satisfaction is associated with the availability and use of medical advice lines,[4] a limitation of our study is the lack of data on the influence of the PAL availability on parents' willingness to take part in the MIS BAIR trial, their satisfaction and retention rates. Future studies should address whether advice lines provide an effective incentive to improve trial participation and/or retention, and seek parents' feedback and evaluation of their benefit. Questionnaires and cluster randomised cohort studies may be of use. Additionally, our results may not be generalisable as the parents, having self-selected to participate in a trial, may not be representative of the wider population. Future PAL services could be improved by having a preplanned referral pathway and information brochure available for common parental concerns.

Paediatric advice services are well recognised as valuable and convenient means of empowering parents and reducing parental concerns, helping to inform decision making and parental help seeking.[4] The MIS BAIR PAL study demonstrates that demand for such a service exists within a clinical trial setting and is likely to be used by parents during early parenthood. Additionally, this prospective cohort study, embedded within a host RCT, adds to the evidence base for trialists conducting a study within a trial, informing design and conduct of future trials.[18]

## Author affiliations

[1]Infectious Diseases Research Group, Murdoch Children's Research Institute, Parkville, Victoria, Australia
[2]The Royal Melbourne Hospital, Parkville, Victoria, Australia
[3]Department of Obstetrics and Gynaecology, Mercy Hospital for Women, Heidelberg, Victoria, Australia
[4]Infectious Diseases Unit, Royal Children's Hospital Research Institute, Parkville, Victoria, Australia
[5]Australian Technical Advisory Group on Immunisation (ATAGI), Melbourne, Victoria, Australia
[6]Vaccine Safety Investigation Group (VSIG), Melbourne, Victoria, Australia
[7]Murdoch Children's Research Institute, Parkville, Victoria, Australia
[8]The Peter Doherty Institute for Infection and Immunity, Melbourne, Victoria, Australia
[9]Laboratory Services, Royal Children's Hospital Research Institute, Parkville, Victoria, Australia
[10]Royal Children's Hospital Research Institute, Parkville, Victoria, Australia
[11]Institute of Infection and Global Health, University of Liverpool, Liverpool, UK
[12]Department of Paediatrics, The University of Melbourne, Parkville, Victoria, Australia
[13]Department of Paediatrics, Mercy Hospital for Women, Heidelberg, Victoria, Australia

**Acknowledgements** The authors thank the MIS BAIR participants and their families for their involvement in the trial. We also thank our recruitment and follow-up teams.

**Collaborators** Submitted on behalf of the Melbourne Infant Study: BCG for Allergy and Infection Reduction trial group.

**Contributors** JC and JK contributed equally to this paper and are joint first authors; transcribed the call data collected, performed statistical analysis and wrote the letter. NC and LFP contributed equally to this paper and are joint last authors, and were integral to the study funding, design and letter production. All other authors were integral in data collection and management of the paediatric advice line, including taking calls and providing advice where necessary, and reviewed and approved the final letter.

**Funding** The Melbourne Infant Study: BCG for Allergy and Infection Reduction trial is funded by research grants from the National Health and Medical Research Council of Australia (GNT1051228 and GNT1099676), The University of Melbourne, The Royal Children's Hospital Foundation, the Murdoch Children's Research Institute, Mercy Health, the Myers Foundation, the European Society of Paediatric Infectious Diseases, the Nossal Institute for Global Health and the Aranday Foundation. LFP is supported by the Swiss National Science Foundation (Early Postdoc Mobility grant number P2GEP3_178155). The guarantor for this study is NC.

**Competing interests** None declared.

**Patient and public involvement** Patients and/or the public were not involved in the design, conduct, reporting or dissemination plans of this research.

**Patient consent for publication** Not applicable.

**Ethics approval** This study involves human participants and the paediatric advice line was included in the Melbourne Infant Study: BCG for Allergy and Infection Reduction ethical and governance approval by Mercy Health Human Research Ethics Committee (HREC, number R12-28) and Royal Children's Hospital HREC (number 33025) with additional governance approval from Barwon Health and St John of God, Geelong, Victoria. Signed informed consent was obtained from the parents of all participants.

**Provenance and peer review** Not commissioned; externally peer reviewed.

**Data availability statement** Data are available in a public, open-access repository.

**ORCID iDs**
Joyce Chan http://orcid.org/0000-0001-8253-8995
Jordan Kirby http://orcid.org/0000-0002-0507-969X
Nigel Curtis http://orcid.org/0000-0003-3446-4594

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
