## [Reviewer comments · BMJ Paediatrics Open]

This paper was submitted to a another journal from Archives of Disease in Childhood but declined for publication following peer review. The authors addressed the reviewers' comments and submitted the revised paper to BMJ Paediatrics Open. The paper was subsequently accepted for publication at BMJ Paediatrics Open.

ARTICLE DETAILS

TITLE (PROVISIONAL)	Use of a paediatric advice line for parents of infants recruited to a randomised controlled trial
AUTHORS	Chan, Joyce Kirby, Jordan Gardiner, Kaya Kareela Abruzzo, Veronica Bannister, Samantha Bellamy, Karen Brophy, Clare Clifford, Vanessa Cox, Lianne Elborough, Hannah Freyne, Bridget Freyne Summons, Freya Morrison, Clare Wall, Kate Casalaz, Daniel M Curtis, Nigel Pittet, Laure F.

VERSION 1 – REVIEW

REVIEWER	Reviewer name: Dr. Chris Gale Institution and Country: Imperial College London, United Kingdom of Great Britain and Northern Ireland Competing interests: None
REVIEW RETURNED	13-May-2022

GENERAL COMMENTS	This letter describes the use of a novel addition - a 'paediatric advice line' - which was provided to participants of a vaccine trial in infancy, MIS-BAIR. The reporting is comprehensive for a letter with a thorough appendix. The study finds that about 1 in 5 families called the PAL within the trial for non-trial related paediatric advice, and that callers were different from non-callers in relation to some parental factors (education, number of other children). I think this is an important addition to paediatric trial methodology and it is important to report these findings. I have a few suggestions that might make the letter more useful to the reader. 1. Can the authors please provide information on the cost of the PAL within the trial and how challenging it was to staff throughout the trial; how many paediatric practitioners were required?2. Can the authors please present the information on recruitment
---

	more clearly in the letter text (currently it is in Appendix B), and discuss it in light of other similar trials - is there any evidence that the PAL led to higher recruitment that would be expected? 3. Can the authors comment on how this PAL could be more robustly tested in a future trial - for example as a Study Within A Trial (SWAT) https://trialsjournal.biomedcentral.com/articles/10.1186/s13063-018-2535-5 Provision of a PAL for example could be cluster randomised to look at the impact on recruitment/retention... Thank you, this was a very interesting study to read; the authors should be commended for writing in such a concise, clear yet comprehensive manner. Chris Gale
--	--

REVIEWER	Reviewer name: Prof. Saul N Faust Institution and Country: University of Southampton, United Kingdom of Great Britain and Northern Ireland Competing interests: None
REVIEW RETURNED	03-Aug-2022

GENERAL COMMENTS	Thank you for asking me to review this letter describing a non-urgent advice line provided (supp material methods) "as recognition for their [parents] contribution to the trial". The description/results describe parents, then conclusion uses "mothers" rather than parents. The authors conclusion is that results show that first time mothers mothers of children born following IVF and more educated mothers are more likely to use the informal helpline. I am not sure what this letter adds to most paediatricians understanding of which parents are more likely to seek advice. The letter may have been more relevant if the authors had focused their conclusions on the fact that an advice line such as the one used appears to add nothing to increasing the diversity of paediatric participation in trials ie the advice line as described appears to be being accessed by those who already use healthcare more effectively or participate in trials. It has not appeared to address increasing participation from people in harder to reach populations, either from lower income/mobility families or diverse communities (English speaking or otherwise). Clearly these are complex issues to tease out involving detailed demographic information about participants and probably formal qualitative interviews with appropriately selected parents in diverse settings – likely to be beyond description in a short letter but that may be able to be considered as part of the ongoing trial.
--

VERSION 3 – REVIEW

REVIEWER	Reviewer name: Dr. Chris Gale Institution and Country: Imperial College London, United Kingdom of Great Britain and Northern Ireland Competing interests: None
REVIEW RETURNED	20-Oct-2022

GENERAL COMMENTS	I had the pleasure of reviewing this manuscript previously. This
--

	updated is much improved and the authors have addressed all reviewer comments effectively. I have no further comments. The authors are to be congratulated on this clearly written and interesting work.
--	--

REVIEWER	Reviewer name: Dr. Peter Flom Institution and Country: Peter Flom Consulting, United States Competing interests: None
REVIEW RETURNED	09-Oct-2022

GENERAL COMMENTS	I confine my remarks to statistical aspects of this paper. The general approach is OK, but I have some issues to resolve before I can recommend publication. GENERAL COMMENTS In various places "multivariate logistic regression" should be replaced with "multiple logistic regression". Multivariate regression means more than one dependent variable. Also, when the authors give a p value, they should also give the test statistic (t, or chi-square or U or whatever). Why wasn't seasonal variation assessed with any statistical tools? SPECIFIC COMMENTS Page 9 - when giving the N for each reason, also give the %. Page 10 - Please describe how the multiple regression model was built. (And here, in the title of the section "multivarite" is OK. I know, the usage is weird, but ... well, that's how the terms are used). From table 2, it looks like bivariate screen was used, but this is not a good method. All the output of the final model will be wrong. p values will be too low, standard errors too small, and parameter estimates biased away from 0 (see Harrell, *Regression Modeling Strategies* for details, proofs, and examples). The best method is substantive knowledge. I like Figure 2. Innovative graphics, and a good one. But it would be better to mark the x access in years or months, rather than weeks. (People don't think of a baby as being 105 weeks old). Also, I think the "Not documented" row could be deleted. However, I do not like Figure 4. Stacked bar charts are not a good graphic (see the work of William S. Cleveland). A mosaic plot might be good here.
--

VERSION 3 - AUTHOR RESPONSE

Author's response to the reviewer's comments

Thank you for taking the time to review our manuscript, submitted as a Research Letter, titled "Use of a paediatric advice line for parents of infants recruited to a randomised controlled trial".

Given we have been asked to revise and re-submit as an Original Article, we have thoroughly expanded on each section, allowing a much more elaborate explanation of the PAL development and results. We have provided a broad overview of the PAL costs, logistics and outcomes. We have

focused on this in both our methods and discussion section, exploring ways of improving future studies. As requested by reviewer 1, we have further explored the consort diagram (Figure 1) in our methods section, providing the source of the original MIS BAIR eligibility criteria. In our discussion, we have referenced the Study Within A Trial (SWAT) information and explored areas of improvement using this design for other trialists.

In response to reviewer 2, we have revised the wording of “parents” and “mothers” in our manuscript. Our findings were that first-time parents, inclusive of both mothers and fathers, were more likely to use the PAL. Mothers with higher educational attainment were also more likely to use the PAL. However, we did not have paternal educational attainment information in our study. Therefore, we reference “first-time parents” and “mother’s educational attainment” as opposed to parental educational attainment.

In our study, we identify sociodemographic factors that were associated with being more likely to use the PAL. This indicates potential barriers to accessing and using the PAL, such as lower educational attainment or lower socio-economic status. We agree with reviewer 2 that this needs further investigation and we suggest that future studies need to focus on both enablers and barriers to PAL use to improve trial methodology, patient satisfaction and health outcomes.

As little data exists in this area of research, we hope that both our reviewers the BMJ Paediatrics Open will see use of the data presented and discussed in this manuscript as both novel and helpful in guiding future studies.

VERSION 4 – REVIEW

REVIEWER	Reviewer name: Dr. Peter Flom Institution and Country: Peter Flom Consulting, United States Competing interests: None
REVIEW RETURNED	04-Nov-2022
GENERAL COMMENTS	The authors have addressed my concerns and I now recommend publication